# Ten Questions on Using Lung Ultrasonography to Diagnose and Manage Pneumonia in the Hospital-at-Home Model: Part I—Techniques and Patterns

**DOI:** 10.3390/diagnostics14242799

**Published:** 2024-12-13

**Authors:** Nin-Chieh Hsu, Yu-Feng Lin, Hung-Bin Tsai, Tung-Yun Huang, Chia-Hao Hsu

**Affiliations:** 1Department of Internal Medicine, College of Medicine, National Taiwan University, Taipei 100225, Taiwan; chesthsu@gmail.com (N.-C.H.); dr.yufenglin@gmail.com (Y.-F.L.); 2Division of Hospital Medicine, Department of Internal Medicine, Taipei City Hospital Zhongxing Branch, Taipei 103212, Taiwan; hbtsai37@gmail.com; 3Taiwan Association of Hospital Medicine, Taipei 100225, Taiwan; 4Department of Otolaryngology, National Taiwan University Hospital, Taipei 100225, Taiwan; b02401072@ntu.edu.tw; 5Department of Orthopedics, Kaohsiung Medical University Hospital, Kaohsiung 807378, Taiwan; 6College of Medicine, Kaohsiung Medical University, Kaohsiung 807378, Taiwan

**Keywords:** pneumonia, point-of-care, ultrasonography, hospital-at-home, diagnosis, treatment, consolidation, air bronchogram

## Abstract

The hospital-at-home (HaH) model delivers hospital-level acute care, including diagnostics, monitoring, and treatments, in a patient’s home. It is particularly effective for managing conditions such as pneumonia. Point-of-care ultrasonography (PoCUS) is a key diagnostic tool in the HaH model, and it often serves as a substitute for imaging-based diagnosis in the HaH setting. Both standard and handheld ultrasound equipment are suitable for lung ultrasound (LUS) evaluation. Curvelinear and linear probes are typically used. Patient positioning depends on their clinical condition and specific diagnostic protocols. To enhance sensitivity, we recommend using at least 10-point protocols supported by studies for pneumonia. Five essential LUS patterns should be identified, including A-line, multiple B-lines (alveolar-interstitial syndrome), confluent B-lines, subpleural consolidation, and consolidation with air bronchogram. Pleural effusion is common, and its internal echogenicity can indicate severity and the need for invasive procedures. The current evidence on various etiologies and types of pneumonia is limited, but LUS demonstrates good sensitivity in detecting abnormal sonographic patterns in atypical pneumonia, tuberculosis, and ventilator-associated pneumonia. Further LUS studies in the HaH setting are required to validate and generalize the findings.

## 1. Introduction

The hospital-at-home (HaH) model is a healthcare approach where patients receive acute care at home, rather than being admitted to a hospital [1,2]. It is designed to provide hospital-level services, including diagnostics, monitoring, treatments, and sometimes even advanced therapies, in a patient’s home environment [3]. This model is especially beneficial for managing conditions like pneumonia, heart failure, chronic obstructive pulmonary disease (COPD), urinary tract infection, and skin and soft tissue infection [3,4,5]. In the HaH model, patients are carefully selected based on their medical stability, condition severity, and the availability of home support systems. Studies show comparable or better outcomes for certain conditions, with fewer complications and higher recovery rates.

Point-of-care service development is essential for delivering the HaH model. These services include the use of portable diagnostics (e.g., point-of-care testing and imaging), remote monitoring, intravenous therapies, and telemedicine consultations [5,6]. While treatments in the hospital-at-home (HaH) model are often comparable to those provided in hospitals, the diagnostic tools are typically novel and rapidly evolving. Point-of-care ultrasonography (PoCUS) is one of the most prominent diagnostic tools in this model [7]. In the hospital setting, the role of PoCUS is primarily for screening or as a complementary examination to support other imaging modalities, such as X-ray or computed tomography [8]. In the HaH setting, however, PoCUS is often regarded as a substitute—and in many cases, the only option—for imaging-based diagnosis [7,9]. This is why PoCUS is increasingly being used to diagnose conditions for which it has not traditionally been the primary diagnostic tool.

Pneumonia is one of the most common infectious diseases in the community [10]. Following the COVID-19 pandemic, many patients have experienced receiving home-based treatment for coronavirus-related pneumonia [11]. Reports in the literature on the use of PoCUS for diagnosing and managing COVID-19 pneumonia are compelling for physicians aiming to treat pneumonia patients in a similar home-based manner [12,13]. In this review, we aim to explore and address several key questions regarding the use of PoCUS for diagnosing and treating pneumonia in the HaH setting.

In this review, we address ten key questions that are essential for the effective use of PoCUS in evaluating and diagnosing pneumonia in HaH practice (Table 1). Based on the current evidence and our experiences, we aim for the answers to and discussions of these questions to be helpful for clinicians in practice.

## 2. Question 1: What Ultrasound Techniques Are Essential for Diagnosing Pneumonia?

Thoracic ultrasonography has been developed by pulmonology and critical care medicine since the 1990s and has shown promise in ultrasound-assisted procedures [14,15,16]. In the 2000s, lung ultrasound (LUS) emerged as a new term, representing a point-of-care tool for evaluation and diagnosis in thoracic medicine [17,18,19,20]. The associated researchers have made significant efforts to dispel the myth that ultrasonographic examination of the lung is impossible [21], ushering in a new paradigm and era. By interpreting various artifacts generated by normal or pathological lungs, LUS enables highly sensitive diagnoses, even in cases of minor lung abnormalities [22].

### 2.1. Ultrasound Equipment

The evaluation of the lung can be performed with standard or handheld ultrasound equipment [23,24,25]. Almost all modern ultrasound equipment includes both B-mode and M-mode, which are essential for lung ultrasound scanning. However, there are limited data available to define optimal scanner settings for the enhanced visualization of lung lesions and artifacts. It should be noted that quantitative comparisons between different equipment are not meaningful for diagnostic purposes. Since the sensitivity of artifacts depends heavily on the equipment and scanning settings [26], there is a call to use the same equipment consistently and standardize scanning protocols to effectively monitor the evolution of artifacts in patients [27]. Color Doppler imaging, while not essential for diagnosis, can be valuable for more detailed assessments of severity and prognosis in patients with pneumonia [27,28,29].

### 2.2. Ultraosund Probes

In adult patients, curvelinear (convex) and linear probes are typically used [27], with the linear probe being particularly effective for observing breath-dependent motion, pleural line abnormalities, and diaphragm function [27,30,31]. The microconvex probe was one of the earliest probes developed for use in the intensive care setting, establishing what is now referred to as the “one probe for all” paradigm [32,33]. In pediatric cases, microconvex probes can be helpful due to theirsmaller footprint and high accuracy [34], however, linear probes are generally preferred due to their ability to provide finer detail, which is beneficial for assessing children’s chest structure [27,34].

The sector array probe (commonly referred to as a “phased array”), like the curvilinear probe, operates at lower frequencies, making it well suited for evaluating deeper structures, such as in cases of pulmonary edema or consolidation. Notably, curvilinear probes may offer higher interpretation accuracy compared to phased array probes, particularly for novice users interpreting pleural pathologies [35]. In the PoCUS approach, utilizing multiple probes for a focused multi-organ and multi-system examination is often necessary. When evaluating patients with suspected pneumonia, a multi-probe approach can provide valuable information about co-morbidities, such as heart failure or venous thrombosis, aiding in severity assessment and triage.

### 2.3. Patient Positioning

Patient positioning depends on their clinical condition and specific diagnostic requirements [27]. For patients in a sitting position, the dorsal lung regions are typically examined [36], while the supine position is used for the anterior regions. Pleural effusion is more easily detected with the patient seated [37], whereas the supine position is better suited for detecting pneumothorax [38,39]. With the patient’s arm raised above the head or positioned with their hand resting on the opposite shoulder, the intercostal spaces widen, allowing access to the subscapular region.

Notably, patient positioning significantly influences the appearance of vertical artifacts observed during LUS. A consistent approach to patient positioning during LUS is essential for accurately monitoring dynamic changes in heart failure [40].

### 2.4. Regions of Examination

LUS may involve either a comprehensive assessment of the entire lung or a targeted investigation of specific areas based on clinical findings [41,42]. Focal examinations are conducted using various scan orientations (longitudinal, transverse, and oblique) and may involve different probes. Typically, the scan starts in symptomatic areas, such as where pleuritic pain is reported or where abnormal breathing sounds are identified on auscultation. In cases where the patient must remain in a supine position, posterior regions can be assessed with the patient in a lateral decubitus position or through a posterolateral approach [17,18]. The apex of the lung can also be assessed using suprasternal, supraclavicular, and infraclavicular approaches. In children, anatomical features such as a thinner chest wall and reduced thoracic width allow for clearer ultrasound imaging [43]. providing high-quality lung images. For non-cooperative patients, the procedure may take longer but remains generally feasible [44].

Recent studies have reported that a focused examination, such as an 8-point LUS protocol, is time-efficient while maintaining similar reproducibility compared to a comprehensive protocol for patients with heart failure [42]. Several targeted protocols were explored during the COVID-19 pandemic using 8-point, 10-point, or 12-zone approaches [13,45,46,47]. However, the use of heterogeneous LUS protocols across departments limits the strength of the evidence level of LUS examination and highlights the need for well-designed trials and standardized methodologies [48].

For the evaluation and diagnosis of patients with pneumonia, earlier reports use a systematic examination of all intercostal spaces in both sitting and supine positions [49]. However, this approach is time-consuming. During the COVID-19 pandemic, focused LUS protocols were frequently proposed. A pilot report from 12 patients with COVID-19 found that all of the patients had diffuse B-lines with spared areas [50]. Although LUS patterns may be nonspecific, the observation of some aspects of vertical artifacts can enhance the diagnostic power, and combining sonographic patterns with blood exam results can aid in accurately characterizing disease phenotypes, supporting effective triage and admission for COVID-19 [51].

Figure 1 illustrates various PoCUS examination regions identified in previous landmark studies. The traditional approach is based on the BLUE protocol, which divides each hemithorax into four areas, creating an 8-point examination framework [17]. The strength of this protocol lies in its evidence-based approach, utilizing conjugated ultrasonic patterns known as “profiles” to assist in differential diagnoses [17]. Unlike other protocols developed later, the BLUE protocol offers a sophisticated LUS-guided approach to clinical reasoning. However, a key limitation of the BLUE protocol is its lower specificity in less-severe patients, with approximately 50% negative findings across the entire protocol when applied to cases of undifferentiated dyspnea in the emergency room setting [52]. Among patients with negative findings from the BLUE protocol, a significant number were ultimately diagnosed with pneumonia [52]. This report suggests that diagnosing pneumonia in less-severe patients, as often encountered in HaH practice, may require examining more points or areas to enhance sensitivity.

The 8-point protocol, derived from the BLUE protocol, is widely adopted in studies [44,53,54]. Among them, a modified BLUE protocol using a new M-point between the superior BLUE point and the diaphragm is used in order to standardize the PLAPS point [55]. Using this additional M-point may increase the sensitivity of pneumonia detection at the right middle lobe.

The 8-point protocol originating from the BLUE protocol is generally consistent in studies [56]. In contrast, 10-point protocols are more heterogeneous, and dorsal-lateral examinations [57], particularly during the COVID-19 pandemic, are less practical due to their low sensitivity without anterior chest assessments. Therefore, we recommend using at least 10-point protocols supported by multiple studies for pneumonia [45,58]. For increased sensitivity, a 12-zone protocol has been proposed for patients with COVID-19 [59,60,61]. The 12-zone protocol is generally consistent, comprising 4 zones derived from the BLUE protocol and 2 additional zones for the dorsal chest wall [62,63].

Laursen et al. developed a 14-zone protocol, combined with focused sonography of the heart and deep veins, which demonstrated broad diagnostic utility for patients with diverse respiratory symptoms [64]. Following this landmark study, several similar 14-zone protocols have been developed in recent studies [65,66].

A 28-point protocol, including 6 points for the dorsal areas, 3 for the lateral areas, and 4 for the anterior chest wall on each side, was reported as a comprehensive examination for COVID-19 pneumonia [67]. However, its accuracy for patients with non-COVID-19 pneumonia remains insufficiently reported. Additionally, scoring each point can be time-consuming, making it less practical for use in the HaH setting.

## 3. Question 2: What Are the Ultrasound Patterns Associated with Pneumonia?

Performing LUS requires less technical skill than abdominal or cardiac ultrasound, with a faster learning curve for beginners to recognize pulmonary interstitial syndrome and consolidation [68,69].

Researchers in the 1990s made a significant breakthrough and revealed typical artifacts observable when examining the pleural surface. Initially, patients with pulmonary sarcoidosis exhibited various pleural abnormalities, characterized by irregular and rough surfaces with coarse interruptions and an increase in artifacts [70]. Subsequently, a study reported that 88.8% of patients with pneumonia presented with consolidation visible on sonograms [71]. Key sonographic artifacts were named by subsequent contributors systematically with validated trials [72,73,74,75,76]. These LUS signs are well known and essential for sonographic diagnosis.

### 3.1. Essential Lung Ultrasound Patterns

Figure 2 illustrates five essential LUS patterns that should be identified. The A-lines represent a normal and full aeration pattern, appearing as parallel curvilinear lines originating from the pleural edge. These lines are caused by the reverberation artifact of the pleura, indicating that the ultrasound beam does not penetrate below the pleural surface [22,73]. To generate A-lines, the ultrasound probe should be angled so that the ultrasound beam is perpendicular to the pleura. The presence of A-lines provides reassurance to sonographers that the lung area being examined is normal. Notably, an irregular pleural surface will not generate A-lines because the ultrasound beam scatters in a non-specular manner.

B-lines are echogenic vertical artifacts extending from the pleural edge to the bottom of the viewing panel (Figure 2), indicating engorged interlobular septa that enable some penetration of the ultrasound beam below the visceral pleura [22,73]. The presence of less than three B-lines in a given intercostal space can be physiological, caused by normal fluid and lymph within the interlobular septa. However, the presence of multiple B-lines or confluent B-lines is considered pathognomonic [72,73,77]. It is most commonly seen in patients with interstitial lung edema, as well as pneumonia [17]. Although B-lines are sensitive artifacts, the specificity of B-lines for pneumonia is low. In a large multicenter study of LUS for the diagnosis and follow-up of pneumonia, B-lines were not considered a LUS finding for pneumonia [49]. Furthermore, a recent study showed that A-lines are significantly associated with the absence of pneumonia, whereas there are no associations between B-lines and pneumonia [78]. It is important to emphasize that in the BLUE protocol, B-lines are combined with other LUS findings to reach a diagnosis, rather than being used in isolation [17]. During the COVID-19 pandemic, B-lines were widely used for screening and were found to correlate with ground-glass opacities (GGO) observed on computed tomography [60,65,79]. However, it should be noted that the high prevalence of GGO patterns is no longer observed in non-pandemic periods, reducing the diagnostic accuracy of B-lines for pneumonia.

Therefore, except in specific situations such as the COVID-19 pandemic or critical care settings with a pre-post dynamic change, B-lines should be considered within a broader differential diagnosis, including any pathology that may alter the subpleural interlobular septa.

Consolidation appears as an echo-poor region, characterized by hypoechoic, wedge-shaped tissue with poorly defined margins [80]. Consolidation patterns occur due to the loss of lung aeration and are not merely artifacts (Figure 2). Within the consolidated area, air-filled bronchi may be visualized as hyperechoic punctiform images, corresponding to air bronchograms (Figure 2) [81]. The presence of a consolidation pattern may represent pneumonia or atelectasis. The dynamic movement of hyperechoic air densities within consolidation, known as dynamic air bronchograms, enhances diagnostic confidence for pneumonia, with a specificity of 94% and a positive predictive value of 97% [76].

The fluid bronchogram represents exudate-packed conducting airways [82]. It occurs less frequently than air bronchograms. The fluid bronchogram is characterized by echo-free tubular structures along the airways and can be differentiated from pulmonary vessels using color Doppler imaging. The bronchial wall is echoic, and more echoic than the vessel wall, whereas the fluid in the segmental bronchi is hypoechoic. The fluid bronchogram reflects exudate-packed conducting airways, a consequence of bronchial obstruction by secretion or a tumor, which arouses suspicion of poststenotic pneumonitis [83].

### 3.2. Pleural Effusion

Thoracic ultrasound has proven to be an effective tool for visualizing pleural effusions. It is particularly advantageous in detecting trace effusions that may be missed by chest X-rays [84]. Additionally, LUS can aid in differentiating the causes of an elevated hemidiaphragm, such as by distinguishing between subpulmonic or subphrenic collections and diaphragmatic paralysis [85,86]. In the context of pneumonia, LUS can provide critical insights into the nature of associated pleural effusions (Figure 3). Based on internal echogenicity, effusions can be categorized as anechoic, complex non-septated, complex septated, or homogeneously echogenic [87]. Early research by Yang et al. [87] has shown that effusions with appearances other than anechoic are always exudative. Anechoic effusions, on the other hand, may represent either transudates or exudates. The presence of pleural septations is particularly significant, as it may predict the need for intrapleural fibrinolytic therapy or surgical intervention in cases of empyema (Figure 3) [88].

We summarize the sonographic findings of community-acquired pneumonia in the current literature in Table 2 [49,71,89,90,91,92,93,94]. Studies that evaluated all intercostal spaces of the thorax have found that using consolidation as a diagnostic marker for pneumonia offers higher specificity compared to the use of B-lines. However, in 8-point and 10-point protocols, relying solely on consolidation may result in lower sensitivity, unless B-lines are also included as a diagnostic LUS pattern. Importantly, normal LUS findings are uncommon in patients with pneumonia, underscoring the strong negative predictive value of LUS in the diagnosis of pneumonia.

## 4. Question 3: Do Different Settings or Etiologies of Pneumonia Influence the Diagnostic Accuracy of Ultrasonography?

Expanding hospital practices to the HaH setting is often a concern, as differences in settings and etiologies can impact the presentation, pathogenesis, diagnosis, treatment response, and prognosis of patients with pneumonia. If HaH treatment is initiated directly from the emergency room or at home, it is reasonable to assume a disease pattern similar to community-acquired pneumonia. In such cases, existing evidence on LUS diagnosis for community-acquired pneumonia may prove useful [90,91,92,93,94]. Special consideration is required for the HaH program to support early hospital discharge [95]. The concerns are twofold. First, they pertain to the use of LUS for monitoring pneumonia treatment response rather than for initial diagnosis. In the subsequent parts of this review, we will present the current evidence and recommended practices.

Second, there is potential for encountering a subset of hospital-acquired pneumonia in this setting. However, current evidence is relatively scare. In critically ill patients, LUS has demonstrated good rule-in and rule-out ability for ventilator-associated pneumonia [96]. If consolidation with either static or dynamic air bronchograms is absent, VAP can be safely excluded. This suggests that LUS remains sensitive for detecting pneumonia caused by hospital-acquired pathogens, despite potential variations in presentation and lobar involvement.

LUS originally developed in the critical care setting has described four main sonographic patterns in landmark studies [17,32]. These criteria included the following: (1) A/B profile: a mix of A-lines and B-lines observed at the anterior four points; (2) B’ profile: B-lines at the anterior four points with absent normal pleural sliding; (3) C profiles: consolidation detected at any of the anterior four points; and (4) A-noV-PLAPS: a profile at the anterior four points combined with no deep-vein thrombosis and positive PLAPS. The combination of these four criteria demonstrated a sensitivity of 89% and a specificity of 94% for diagnosing pneumonia [32]. However, clinicians rarely encounter patients with severe hypoxemia in the HaH setting, unless the patient is terminally ill and receiving palliative care. Consequently, the specificity of pneumonia diagnosis based on the four criteria may be lower than anticipated.

In the HaH setting, clinicians may encounter patients who use home ventilators. Using a 12-zone protocol for patients with mechanical ventilation in the hospital, the sensitivity, specificity, and diagnostic accuracy for lung ultrasound were 100, 78, and 95% for consolidation, 94, 93, and 94% for interstitial syndrome using computed tomography as the gold standard, respectivley [97]. This suggests that LUS findings are not substantially influenced by positive pressure ventilation and can reliably screen and diagnose VAP in patients using home ventilators.

LUS may also assist in diagnosing atypical pneumonia, even though atypical pathogens are typically covered by empiric antibiotics for community-acquired pneumonia. A recent pooled analysis of four studies, including a total of 274 patients with a diagnosis of atypical pneumonia (most of which was Mycoplasma pneumonia), reported that both consolidation (84–100%) and B-lines (85%) were common [98]. Atypical pneumonia presented with consolidation associated with subpleural effusion and a diffuse interstitial pattern. Although both scattered and confluent B-lines were described, the B-lines were confluent in more-severe disease.

A pooled analysis of seven studies for patients with pulmonary tuberculosis found that circular or ellipsoidal hypoechoic subpleural lesions, generally <1.5 cm, were the most common finding by LUS. These so-called “subpleural nodes” were reported in up to 90% of adults, mainly in the superior quadrants of the lung [98]. Therefore, identifying an isolated subpleural consolidation in the upper lung should raise suspicion of tuberculosis in patients presenting with pneumonia-like symptoms.

## 5. Conclusions

Clinicians seeking to diagnose and manage pneumonia within a hospital-at-home model must first develop the ability to identify sonographic patterns of pneumonia, utilizing appropriate equipment and techniques. In the first part of our review, we summarize the current point-of-care ultrasound practices for diagnosing pneumonia in ward and emergency department settings. While evidence supporting the implementation of similar practices outside of the hospital is limited and requires further investigation, understanding the strengths and limitations of these approaches provides valuable insights for developing effective models in HaH programs. In the subsequent parts of our review, we will address differential diagnoses, potential pitfalls and confounders, and follow-up considerations regarding LUS in pneumonia.

## Figures and Tables

**Figure 1 diagnostics-14-02799-f001:**
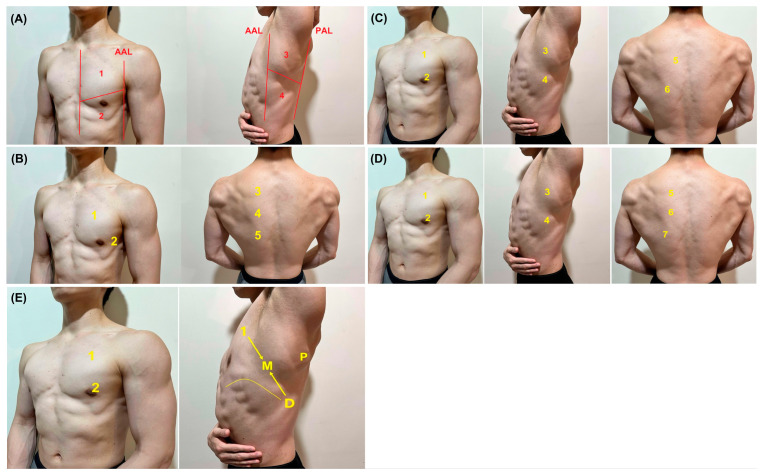
The (**A**) 8-point protocol (Point 1–4 for left hemithorax), (**B**) 10-point protocol (Point 1–5 for left hemithorax, (**C**) 12-zone protocol (Point 1–6 for left hemithorax), and (**D**) 14-zone protocols (Point 1–7 for left hemithorax) for the sonographic diagnosis of pneumonia in landmark studies, and the (**E**) modified BLUE protocol (1, superior BLUE point; 2, inferior BLUE point; D, diaphragm point; M, M-point; P, posterolateral point).

**Figure 2 diagnostics-14-02799-f002:**
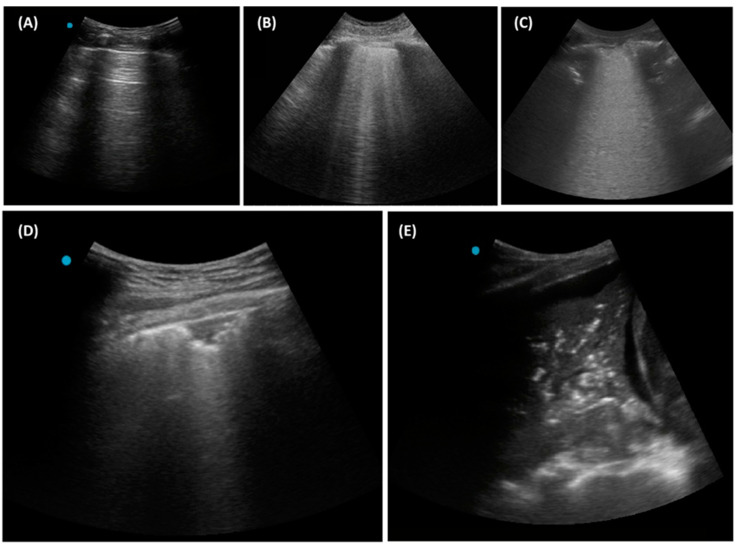
(**A**) A-line; (**B**) Multiple non-confluent B-lines; (**C**) Confluent B-lines; (**D**) Subpleural consolidation; (**E**) Large consolidation with air bronchogram.

**Figure 3 diagnostics-14-02799-f003:**
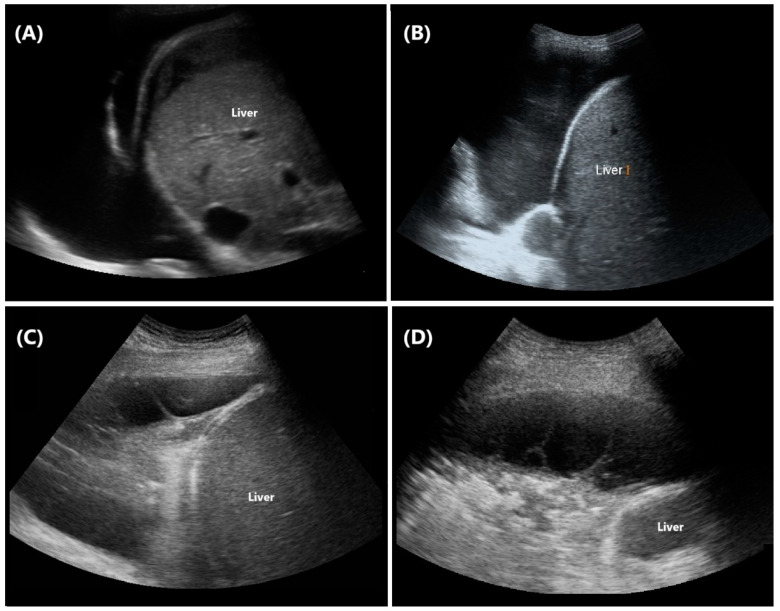
Internal echogenicity of pleural effusion. (**A**) Anechoic: entirely echo-free fluid, (**B**) Complex non-septated: heterogeneously hyperechoic spots within the effusion, also referred to as the “Plankton sign”, (**C**) Complex septated: presence of visible septa or fibrin strands, and (**D**) Loculated: effusion confined between the parietal and visceral pleura with sharply defined margins.

**Table 1 diagnostics-14-02799-t001:** Ten essential questions to address before using point-of-care ultrasonography to diagnose and manage pneumonia in the hospital-at-home model.

What ultrasound techniques are essential for diagnosing pneumonia?
2. What are the ultrasound patterns associated with pneumonia?
3. Do different settings or etiologies of pneumonia influence the diagnostic accuracy of ultrasonography?
4. Do pulmonary comorbidities affect the accuracy of ultrasound diagnosis of pneumonia?
5. Do other differential diagnoses mimic the ultrasound patterns of pneumonia?
6. Do ultrasound findings correlate with pneumonia severity?
7. Do initial ultrasound findings associated with pneumonia hold prognostic value?
8. Do the ultrasound patterns improve in accordance with pneumonia recovery?
9. Is ultrasound superior to chest X-ray for diagnosing pneumonia?
10. Does ultrasonography lead to the overdiagnosis of pneumonia?

**Table 2 diagnostics-14-02799-t002:** Sonographic findings in patients with non-COVID-19 community-acquired pneumonia.

	Gehmacher, 1995 [71] (*n* = 143)	Reissig, 2007 [89] (*n* = 33)	Parlamento, 2009 [90] (*n* = 49)	Sperandeo, 2011 [91] (*n* = 342)	Reissig, 2012 [49] (*n* = 362)	Cortellaro, 2012 [92] (*n* = 80)	Liu, 2015 [93] (*n* = 112)	Pagano, 2015 [94] (*n* = 68)
Setting	N/A	Ward	ED	Ward	Ward	ED	ED	ED
Participants	Adult CAP	Adult CAP	Adult CAP	Adult CAP	Adult CAP	Adult CAP	Adult CAP	Adult CAP
Zones	N/A	All ICS	10 zones	All ICS	All ICS	10 zones	All ICS	8 zones
US findings								
Consolidation	88.8%	100%	96.9%	92%	97.6%	91.3%	71.4%	70.6%
B-lines (AIS)	N/A	N/A	68.8%	N/A	N/A	61.3%	39.3%	27.9%
Air bronchogram	88.1%	97%	50.0%	70%	86.7%	88.8%	N/A	N/A
Fluid bronchogram	N/A	0%	N/A	31%	8.1%	35%	N/A	N/A
Pleural effusion	54.5%	70%	34.4%	35.1%	54.4%	38.8%	N/A	4.4%
Normal LUS	11.2%	0%	0%	8%	0%	0%	N/A	1.5%

Abbreviations: CAP, community-acquired pneumonia; ED, emergency department; ICS, intercostal space; N/A, not available.

## Data Availability

Not applicable.

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
