# Peer review of "Ten Questions on Using Lung Ultrasonography to Diagnose and Manage Pneumonia in the Hospital-at-Home Model: Part I—Techniques and Patterns"

_diagnostics, 2024, doi:10.3390/diagnostics14242799_

Round 1
Reviewer 1 Report
Comments and Suggestions for Authors
I thank the authors for the opportunity they have given me to read this interesting review.
The authors present an updated review on Using Lung Ultrasonography to Diagnose and Manage Pneumonia in Hospital-at-Home Model. They do so by answering 10 questions, 5 of which are asked in this paper, and it seems that the other 5 will be answered in a second paper.
The authors have made an interesting update on the subject and the data they present are up to date.
I think this is a very interesting review for healthcare professionals who manage patients in home hospitalisation.
The points that have been addressed are very interesting, well developed and cover everything that the professional should know in a basic way.
The bibliography is well selected.
I think that the review is very accurate and with a second paper that completes the missing information, it could be a very complete review of the subject.
Author Response
# Reviewer #1:
I thank the authors for the opportunity they have given me to read this interesting review.
Comment 1: The authors present an updated review on Using Lung Ultrasonography to Diagnose and Manage Pneumonia in Hospital-at-Home Model. They do so by answering 10 questions, 5 of which are asked in this paper, and it seems that the other 5 will be answered in a second paper.
The authors have made an interesting update on the subject and the data they present are up to date.
I think this is a very interesting review for healthcare professionals who manage patients in home hospitalisation.
The points that have been addressed are very interesting, well developed and cover everything that the professional should know in a basic way.
The bibliography is well selected.
I think that the review is very accurate and with a second paper that completes the missing information, it could be a very complete review of the subject.
Response 1: Thank you for your positive comments on our work.
Reviewer 2 Report
Comments and Suggestions for Authors
This is a narrative review of the use of lung POCUS to screen for pneumonia. Although this review is targeted toward utilization of POCUS within the hospital-at-home (HaH) model, the review is broadly applicable to use of POCUS to screen for pneumonia in any context. This review is well written and covers important concepts that are of relevance to anyone using POCUS to screen for pneumonia.
However, there are a few (relatively minor) and easily fixable issues that could be addressed to improve the manuscript further.
1) Line 101: “The phased array probe, like the curvilinear, operates at lower frequencies…”
Unbeknownst to most POCUS practitioners, the term “phased array probe” is misleading because all modern ultrasound transducers utilize phased-array technology, including even linear high-frequency probes. So all modern ultrasound transducers are “phased array” and this name does not uniquely describe the probe in question. A more technically accurate
Unbeknown to many POCUS practitioners, the term “phased-array probe” is actually a colloquial term and a misleading one at that because electronic phasing is done by the ultrasound machine (not in the probe itself) and occurs for all modern ultrasound transducers (i.e., even linear high-frequency probes are “phased array”). So it is the ultrasound system that is “phased” and each modern transducer benefits from this electronic “phasing.” So the technical term for the probe in question (colloquially called the “phased-array probe”) is “linear sector arc phased array probe”. But again since all modern ultrasound transducers use phasing, reasonable simplified names for this probe are any of the following: sector, sector arc, or sector array. These terms are actually used widely in the Radiology and ultrasound physics literature to describe this probe, but seem to have been overlooked by point-of-care ultrasound providers for some reason. I would revise the sentence to something like the following:
“The sector array probe (colloquially called “phased array”), like the curvilinear, operates at lower frequencies…”
Here are some relevant references in case the authors are interested. Background of how phased array ultrasound systems were developed:
i) vonRamm, O. T. & Thurstone, F. L. Cardiac imaging using a phased array ultrasound system. I. System design. Circulation. 53 (2), 258-262, (1976).
ii) Kisslo, J., vonRamm, O. T. & Thurstone, F. L. Cardiac imaging using a phased array ultrasound system. II. Clinical technique and application. Circulation. 53 (2), 262-267, (1976).
Examples of the term “sector-array probe” and “sector probe” in the radiology and ultrasound physics literature:
iii) Smit MR, de Vos J, Pisani L, Hagens LA, Almondo C, Heijnen NFL, Schnabel RM, van der Horst ICC, Bergmans D, Schultz MJ, Bos LDJ. Comparison of Linear and Sector Array Probe for Handheld Lung Ultrasound in Invasively Ventilated ICU Patients. Ultrasound Med Biol. 2020;46(12):3249-56. PubMed PMID: 32962892.
iv) Tasci O, Hatipoglu ON, Cagli B, Ermis V. Sonography of the chest using linear-array versus sector transducers: Correlation with auscultation, chest radiography, and computed tomography. J Clin Ultrasound. 2016;44(6):383-9. PubMed PMID: 26863904.
2) Lines 163-167: Figure 1 would have greater educational value if you added an additional panel demonstrating this modified BLUE protocol with the additional M-point.
3) Figure 2: I think the legend for this figure is a bit confusing. Specifically, panel B shows multiple NON-confluent B-lines and describes this as “alveolar-interstitial syndrome” whereas panel C shows confluent B-lines and includes no descriptor of this finding. I think a more standard interpretation would be: (B) multiple non-confluent B-lines = likely to be primarily interstitial edema whereas (C) confluent B-lines typically implies interstitial-alveolar edema (i.e., a more advanced form of edema where fluid is filling the alveoli in addition to the interstitial spaces).
Author Response
# Reviewer #2
This is a narrative review of the use of lung POCUS to screen for pneumonia. Although this review is targeted toward utilization of POCUS within the hospital-at-home (HaH) model, the review is broadly applicable to use of POCUS to screen for pneumonia in any context. This review is well written and covers important concepts that are of relevance to anyone using POCUS to screen for pneumonia.
However, there are a few (relatively minor) and easily fixable issues that could be addressed to improve the manuscript further.
Comment 1: Line 101: “The phased array probe, like the curvilinear, operates at lower frequencies…”
Unbeknownst to most POCUS practitioners, the term “phased array probe” is misleading because all modern ultrasound transducers utilize phased-array technology, including even linear high-frequency probes. So all modern ultrasound transducers are “phased array” and this name does not uniquely describe the probe in question. A more technically accurate
Unbeknown to many POCUS practitioners, the term “phased-array probe” is actually a colloquial term and a misleading one at that because electronic phasing is done by the ultrasound machine (not in the probe itself) and occurs for all modern ultrasound transducers (i.e., even linear high-frequency probes are “phased array”). So it is the ultrasound system that is “phased” and each modern transducer benefits from this electronic “phasing.” So the technical term for the probe in question (colloquially called the “phased-array probe”) is “linear sector arc phased array probe”. But again since all modern ultrasound transducers use phasing, reasonable simplified names for this probe are any of the following: sector, sector arc, or sector array. These terms are actually used widely in the Radiology and ultrasound physics literature to describe this probe, but seem to have been overlooked by point-of-care ultrasound providers for some reason. I would revise the sentence to something like the following:
“The sector array probe (colloquially called “phased array”), like the curvilinear, operates at lower frequencies…”
Here are some relevant references in case the authors are interested. Background of how phased array ultrasound systems were developed:
i) vonRamm, O. T. & Thurstone, F. L. Cardiac imaging using a phased array ultrasound system. I. System design. Circulation. 53 (2), 258-262, (1976).
ii) Kisslo, J., vonRamm, O. T. & Thurstone, F. L. Cardiac imaging using a phased array ultrasound system. II. Clinical technique and application. Circulation. 53 (2), 262-267, (1976).
Examples of the term “sector-array probe” and “sector probe” in the radiology and ultrasound physics literature:
iii) Smit MR, de Vos J, Pisani L, Hagens LA, Almondo C, Heijnen NFL, Schnabel RM, van der Horst ICC, Bergmans D, Schultz MJ, Bos LDJ. Comparison of Linear and Sector Array Probe for Handheld Lung Ultrasound in Invasively Ventilated ICU Patients. Ultrasound Med Biol. 2020;46(12):3249-56. PubMed PMID: 32962892.
iv) Tasci O, Hatipoglu ON, Cagli B, Ermis V. Sonography of the chest using linear-array versus sector transducers: Correlation with auscultation, chest radiography, and computed tomography. J Clin Ultrasound. 2016;44(6):383-9. PubMed PMID: 26863904.
Response 1: Thank you for pointing out a misleading term which has been commonly used to describe sector probe using for the cardiac exam. We are pleased to use “sector array probe” to replace it.
Comment 2: Lines 163-167: Figure 1 would have greater educational value if you added an additional panel demonstrating this modified BLUE protocol with the additional M-point.
Response 2: Yes, we would like to add a new panel (E) for modified BLUE protocol with M-point in Figure 1.
Comment 3: Figure 2: I think the legend for this figure is a bit confusing. Specifically, panel B shows multiple NON-confluent B-lines and describes this as “alveolar-interstitial syndrome” whereas panel C shows confluent B-lines and includes no descriptor of this finding. I think a more standard interpretation would be: (B) multiple non-confluent B-lines = likely to be primarily interstitial edema whereas (C) confluent B-lines typically implies interstitial-alveolar edema (i.e., a more advanced form of edema where fluid is filling the alveoli in addition to the interstitial spaces).
Response 3: Thank you for your suggestions. Since Figure 2 is intended to illustrate five distinct categories of lung ultrasound findings, we will refrain from delving deeply into the underlying pathophysiological mechanisms. Therefore, we would consider revise as: “(B) Multiple non-confluent B-lines; (C) Confluent B-lines.”
We appreciate your suggestions for revisions that improve the readability and accuracy of our review article.